# Correction of a homoplasmic mitochondrial tRNA mutation in patient-derived iPSCs via a mitochondrial base editor

Xiaoxu Chen[1,7], Mingyue Chen[1,7], Yuqing Zhu[2,7], Haifeng Sun [1,7], Yue Wang[3,7], Yuan Xie[4], Lianfu Ji[5], Cheng Wang [4], Zhibin Hu [6], Xuejiang Guo [3], Zhengfeng Xu [2✉], Jun Zhang [1✉], Shiwei Yang [5✉], Dong Liang [2✉] & Bin Shen [1,6✉]

Pathogenic mutations in mitochondrial DNA cause severe and often lethal multi-system symptoms in primary mitochondrial defects. However, effective therapies for these defects are still lacking. Strategies such as employing mitochondrially targeted restriction enzymes or programmable nucleases to shift the ratio of heteroplasmic mutations and allotopic expression of mitochondrial protein-coding genes have limitations in treating mitochondrial homoplasmic mutations, especially in non-coding genes. Here, we conduct a proof of concept study applying a screened DdCBE pair to correct the homoplasmic m.A4300G mutation in induced pluripotent stem cells derived from a patient with hypertrophic cardiomyopathy. We achieve efficient G4300A correction with limited off-target editing, and successfully restore mitochondrial function in corrected induced pluripotent stem cell clones. Our study demonstrates the feasibility of using DdCBE to treat primary mitochondrial defects caused by homoplasmic pathogenic mitochondrial DNA mutations.

[1] State Key Laboratory of Reproductive Medicine and Offspring Health, Women's Hospital of Nanjing Medical University, Nanjing Maternity and Child Health Care Hospital, Gusu School, Nanjing Medical University, Nanjing 211166, China. [2] Department of Prenatal Diagnosis, Women's Hospital of Nanjing Medical University, Nanjing Maternity and Child Health Care Hospital, Nanjing 210004, China. [3] State Key Laboratory of Reproductive Medicine, Department of Histology and Embryology, Nanjing Medical University, Nanjing 211166, China. [4] Department of Bioinformatics, School of Biomedical Engineering and Informatics, Nanjing Medical University, Nanjing 211166, China. [5] Department of Cardiology, Children's Hospital of Nanjing Medical University, Nanjing 210008, China. [6] Department of Epidemiology, Center for Global Health, School of Public Health, Nanjing Medical University, Nanjing 211166, China. [7] These authors contributed equally: Xiaoxu Chen, Mingyue Chen, Yuqing Zhu, Haifeng Sun, Yue Wang. ✉email: zhengfeng_xu_nj@163.com; zhang_jun@njmu.edu.cn; jrdoctoryang@163.com; liangdong@njmu.edu.cn; binshen@njmu.edu.cn

Primary mitochondrial defects (PMDs) are a group of rare, heterogeneous disorders with mitochondrial defects, which are often caused by pathogenic mutations in mitochondrial DNA (mtDNA)[1]. In human cells, the mitochondrial genome is a 16 kb circular double-stranded DNA, encoding 37 genes consisting of 2 ribosomal RNAs (rRNAs), 22 transfer RNAs (tRNAs), and 13 oxidative phosphorylation subunits[2]. Unlike the nuclear genome, the mtDNA is present in hundreds to thousands of copies per cell, and is entirely maternally inherited in human[3]. Previous reports showed that the prevalence of PMDs is not low, with about 12.5 per 100,000 in adults[4] and 4.7 per 100,000 in children[5]. Notably, there are more PMD patients caused by mtDNA mutations than by nuclear mutations, where the majority are due to single nucleotide variants (SNVs) of mtDNA[4]. Up to now, pathogenic variants have been identified in 36 of the 37 mtDNA-encoded genes[6]. The PMDs commonly affect the tissues that require high levels of cellular energy for proper function, such as the cardiac muscle, nervous system, skeletal muscle, and liver, often leading to severe and multi-system symptoms. However, due to the complexity of the underlying mechanism, most PMD patients after genetic diagnoses are often provided with supportive care, such as exercise, vitamin and xenobiotic use, and standard medications for cardiomyopathy and seizures[7]. Currently, although urgently needed in the clinic, effective and clinically validated treatments are still lacking.

Recent advances in the field of gene editing technology have enabled efficient manipulation in the nuclear genome using CRISPR/Cas9 and its derived technologies, but none of them have shown promising potential for gene editing in the mitochondrial genome, due to challenges including the impermeability of the mitochondrial membrane for RNA[8]. Mitochondrially targeted restriction enzymes and programmable nucleases could degrade mutant mtDNA to shift the ratio of heteroplasmic mutations, by taking advantage of the lack of a DNA repair system in mitochondria[9–11]. However, these methods are unable to repair homoplasmic disease-causing mutations, which can be associated with tissue-specific PMDs including LHON, deafness and cardiomyopathy[12]. Allotopic expression of mitochondrial protein-coding genes in nucleus has been developed as a potential approach to treat PMDs[13]. However, this strategy does not work for most mtDNA-encoded proteins, because high hydrophobicity of these allotopically expressed proteins hampers their import into mitochondria, also, the successful functional integration of these proteins into an assembled complex remains a challenge[14]. Moreover, this approach is not effective for mutations in mitochondrial non-coding genes. Hence, new approaches are urgently needed to repair homoplasmic mutations, regardless of whether such mutations are in mitochondrial coding or noncoding genes.

Recently, the use of RNA-free DddA-derived cytosine base editors (DdCBE) promises to enable CRISPR-free mitochondrial base editing[15]. We and other groups have successfully applied the DdCBE to install mtDNA mutations in zebrafish, rat and mouse for modelling mitochondrial diseases[16–19]. In addition, we have evaluated DdCBE in editing the human tripronuclear embryos, showing its feasibility in mediating mtDNA base-editing during early embryogenesis in human[20]. The emergence of DdCBE has made it theoretically possible to correct pathogenic mtDNA mutations, including both homoplasmy and heteroplasmy. However, the effectiveness of DdCBE in correcting pathogenic mtDNA mutations, especially homoplasmic mutations, has yet to be evaluated before any preclinical applications.

Homoplasmic m.A4300G has been documented as a pathogenic mtDNA mutation in 4 publications with 37 affected individuals recorded[21], leading to maternally inherited hypertrophic cardiomyopathy (HCM) and defected function of mitochondrial respiratory chain in cardiac tissue[22]. Here, we conducted a proof-of-concept study using induced pluripotent stem cells (iPSCs) derived from a HCM patient with homoplasmic m.A4300G mutation, and applied DdCBE-mediated base editing to correct this mutation. We achieved efficient correction of this pathogenic mutation with few off-target editing, and recovered the mitochondrial functions correspondingly, demonstrating the feasibility of using DdCBE to treat PMDs with homoplasmic pathogenic mtDNA mutations.

## Results

**Generation of m.A4300G patient-derived iPSCs.** A one-year and three-month-old boy was admitted to the *Children's Hospital of Nanjing Medical University* due to recurrent convulsions and lethargy. The boy exhibited growth retardation, low muscle strength and an inability to stand alone. Echocardiography revealed the features of hypertrophic cardiomyopathy and congestive heart failure (Supplementary Fig. 1a). In detail, echocardiogram showed mild hypertrophy of ventricular septum and left ventricular posterior wall, enlarged left ventricular end-diastolic diameter (LVED, 47 mm), and substantially decreased left ventricular systolic function (LVEF, 29.2%, Supplementary Fig. 1b). Furthermore, laboratory examinations showed a markedly elevated plasma level of B-type natriuretic peptide (BNP 4158 pg/mL, more than 40 times the upper limit of normal) and mild hyperlactatemia (lactic acids 3.1 mmol/L, upper limit of normal is <2.0 mmol/L) (Supplementary Fig. 1b). No pathogenic or likely-pathogenic variants known to be associated with this disease were identified in the patient through whole exome sequencing, while mitochondrial DNA sequencing identified that the boy carried a homoplasmic m.A4300G mutation in mitochondrial tRNA$^{Ile}$ gene (Supplementary Fig. 1c and d). Family history revealed that the child inherited the m.A4300G mutation from his mother, although his mother did not manifest any similar symptoms (Supplementary Fig. 1d and e). Mitochondrial A4300G mutation was reported to be associated with maternally inherited HCM[22,23]. Based on the clinical and genetic characteristics above, the boy was finally diagnosed with mitochondrial cardiomyopathy. With mitochondrial cocktail treatment, his recent echocardiography revealed a stable condition of the heart with no further hypertrophy of interventricular septum. Previous pedigree studies have shown that the illness caused by m.A4300G mutation would have an unfavorable course, with some patients developing progressive heart failure and eventually dying[23]. Therefore, we set out to generate the patient-derived iPSC with homoplasmic m.A4300G mutation (m.A4300G-iPSC) and to develop potential gene therapy strategy on this cell via DdCBE-mediated gene correction.

The m.A4300G-iPSC was generated from the boy's peripheral blood mononuclear cells (PBMCs) by delivering the Yamanaka factors using non-integrative Sendai virus (Fig. 1a and Supplementary Fig. 2a). The obtained iPSC line had a normal karyotype as expected (Fig. 1b) and expressed pluripotent markers (Fig. 1c and Supplementary Fig. 2b). In vitro differentiation studies proved that the m.A4300G-iPSCs have the potential to differentiate into the three germ layers (Supplementary Fig. 2c), as well as cardiomyocytes (Supplementary Fig. 2d and e). In addition, the homoplasmic m.A4300G mutation could be detected in m.A4300G-iPSCs at different passages (Supplementary Fig. 2f). Taken together, we successfully generated iPSCs with the homoplasmic m.A4300G mutation from a HCM patient.

**m.A4300G mutation causing mitochondrial dysfunction in patient-derived iPSCs.** m.A4300G mutation has been reported to decrease the level of mature mitochondrial tRNA$^{Ile}$[22]. Consistently, tRNA-seq analysis revealed the down-regulation of mitochondrial tRNA$^{Ile}$ in our patient-derived iPSCs (Fig. 1d). Quantitative real-time reverse-transcription PCR (qRT-PCR)

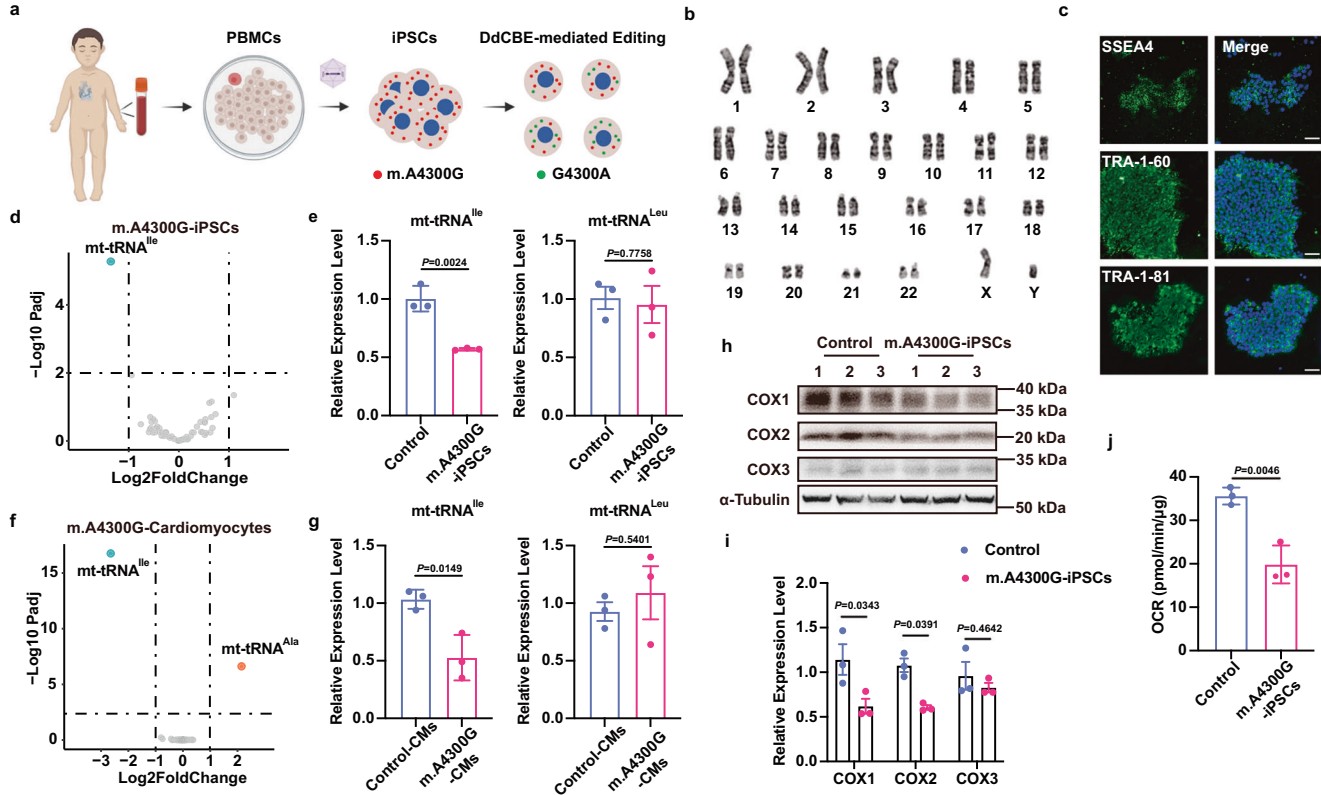

**Fig. 1 m.A4300G mutation causes mitochondrial dysfunction in patient-derived iPSCs. a** Schematic overview of DdCBE-mediated gene correction of m.A4300G mutation in patient-derived iPSCs. Creatd with BioRender.com. **b** The karyotype of patient-derived m.A4300G-iPSCs. **c** Immunofluorescence staining of pluripotency markers SSEA4, TRA-1-60 and TRA-1-81 in m.A4300G-iPSCs. Scale bars, 50 μm. **d** Volcano plot of differentially expressed tRNA genes between control and m.A4300G-iPSCs. Significant genes were selected by fold change (>2- or < −2-fold) and adjusted *P*-value (<0.01). The significantly up-regulated and down-regulated genes are indicated with blue and orange dots, respectively, while non-significant genes are shown as grey dots. **e** qRT-PCR verification of mt-tRNA^Ile (Effect size = −0.4333) and mt-tRNA^Leu (Effect size = −0.0566) expression in control and m.A4300G-iPSCs. Data are presented as mean ± SEM, *n* = 3 independent experiments. **f** Volcano plot of differentially expressed tRNA genes between control and m.A4300G iPSCs-derived cardiomyocytes. **g** qRT-PCR verification of mt-tRNA^Ile (Effect size = −0.4333) and mt-tRNA^Leu (Effect size = −0.0566) expression in control derived cardiomyocytes (Control-CMs) and m.A4300GiPSCs-derived cardiomyocytes (m.A4300G-CMs). Data are presented as mean ± SEM, *n* = 3 independent experiments. **h** Image of Western blot showed the protein level of COX1, COX2 and COX3 in control and m.A4300G iPSCs. **i** The quantitative analysis of COX1 (Effect size = −0.5222), COX2 (Effect size = −0.4720) and COX3 (Effect size = −0.1323) protein level from panel h. Data are presented as mean ± SD, *n* = 3 independent experiments. **j** The quantitative analysis of basal oxygen consumption rate (OCR) by Seahorse assay (Effect size = −15.77). Data are presented as mean ± SD, *n* = 3 independent experiments. Significance was calculated with unpaired two-tailed Student's *t*-test.

results further demonstrated the significantly decreased level of mitochondrial tRNA^Ile and relatively stable level of mitochondrial tRNA^Leu in the patient-derived iPSCs (Fig. 1e). After in vitro differentiation of iPSCs into cardiomyocytes, the steady-state level of mature tRNA^Ile in the m.A4300G-cardiomyocytes was also lower than that in control cardiomyocytes (Fig. 1f, g), indicating that m.A4300G mutation also alters the expression level of mitochondrial tRNA^Ile in cardiomyocytes. Unexpectedly, we observed the up-regulation of mitochondrial tRNA^Ala in m.A4300G-cardiomyocytes (Fig. 1f). Increased alanine level is often found in patients with mitochondrial disorders, because mitochondrial dysfunction may cause more pyruvate accumulation, which can then be converted to alanine by alanine transaminase[24–27]. We speculate that the up-regulation of mitochondrial tRNA^Ala is probably a response to the elevated alanine level in m.A4300G-cardiomyocytes. It has been reported that m.A4300G mutation resulted in the defect of respiratory chain enzymes, such as COX activity[22]. In m.A4300G-iPSCs, we did detect a significant decrease in the levels of COX1 and COX2 proteins, with no significant difference observed in COX3 (Fig. 1h, i). Consistently, the cells showed lower basal oxygen

consumption rates (OCR) compared with control iPSCs (Fig. 1j). Together, m.A4300G mutation leads to a low steady-state level of mature tRNA^Ile and impairs mitochondrial function in the patient-derived iPSCs.

**Correction of m.A4300G mutation in patient-derived iPSCs.**
m.A4300G mutation leads to the arise of a tC motif, which is a preferential context for DdCBE targeting. DdCBEs were reported to induce bystander mutations within spacing region[15]. In order to accurately correct the m.A4300G mutation with minimal bystander mutations, we intended to test the DdCBE pairs with different TALE arrays, different spacing regions and different combinations of the DddA_tox split. We first designed 8 NLS-DdCBE pairs with TALE arrays recognizing the mtDNA heavy (H) or light (L) strand (H1 and H2 or L1 and L2, respectively) and different orientation of the DddA_tox split (G1333N + G1333C, G1333C + G1333N, G1397N + G1397C or G1397C + G1397N), targeting a 13 or 16 bp spacer sequence (Fig. 2a, b). As previously reported[18], we screened each NLS-DdCBE pair by co-transfecting them with the pEGFP-N1 plasmid harboring

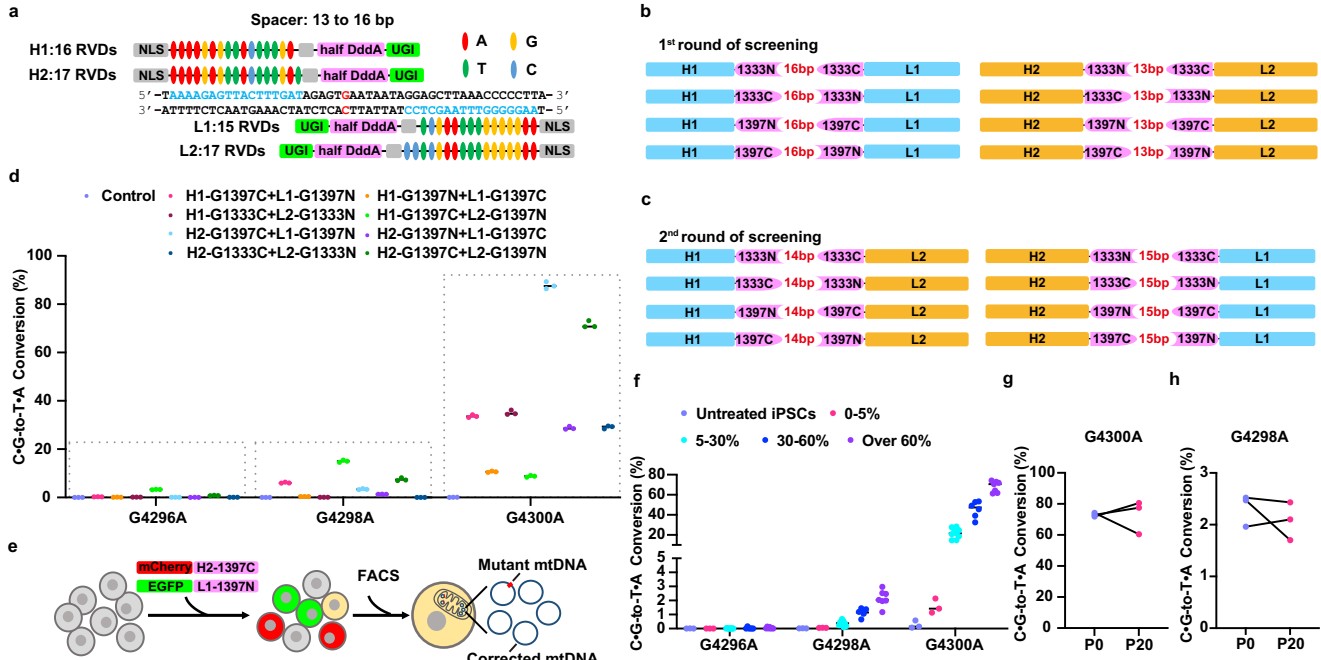

**Fig. 2 Gene correction of m.A4300G mutation in m.A4300G-iPSCs via DdCBE. a** Design of TALE array targeting the mtDNA heavy (H) or light (L) strands (H1 and H2 or L1 and L2, respectively) to target G4300A. G4300 is in red, and TALE binding sequences are in blue. **b** First round screening of H1-DdCBE + L1-DdCBE pair with 16 bp spacing region and H2-DdCBE+L2-DdCBE with 13 bp spacing region. **c** Second round screening of H1-DdCBE + L2-DdCBE pair with 14 bp spacing region and H2-DdCBE+L1-DdCBE pair with 15 bp spacing region. **d** Editing efficiencies of selected DdCBE pairs within spacing region in HEK293FT cells. n = 3 independent experiments for each DdCBE pair. **e** Experimental workflow of DdCBE-mediated gene correction in m.A4300G-iPSCs. **f** Editing efficiencies of H2-G1397C + L1-G1397N DdCBE pair within spacing region in different single-cell A4300G-iPSCs clones. **g** The frequencies of G4300 in passage 0 (P0) and P20 of three corrected iPSC clones. **h** The frequencies of bystander mutation G4298 in P0 and P20 of three corrected iPSC clones.

the m.A4300G target sequence into HEK293FT cells. The target pEGFP-N1 plasmid was recovered and amplified for Sanger sequencing 72 h post-nucleofection. The results showed that four DdCBE pairs (H1-G1397C + L1-G1397N, H1-G1397N + L1-G1397C, H2-G1333C + L2-G1333N, H2-G1397C + L2-G1397N) achieved C·G-to-T·A conversion at G4300 site; among them, H2-G1397C + L2-G1397N pair yielded the highest editing efficiency with over 50% of G being converted to A (Supplementary Fig. 3a). These results encouraged us to perform another round of screening by exchange pairing: H1-DdCBE with L2-DdCBE and H2-DdCBE with L1-DdCBE (Fig. 2c). The Sanger sequencing results showed that the G signal was almost lost at G4300 site in H2-G1397C + L1-G1397N-nucleofected cells, while no obvious bystander mutations were observed within the spacer (Supplementary Fig. 3b). Further deep sequencing analysis revealed that H2-G1397C + L1-G1397N pair yielded the highest editing efficiency at G4300 (87.73 ± 0.85%) compared with other pairs, while maintained a relatively low bystander editing at G4298 (3.42 ± 0.12%) site (Fig. 2d). Therefore, the screened H2-G1397C + L1-G1397N pair was chosen for the following gene therapy experiments in m.A4300G-iPSCs.

The H2-G1397C and L1-G1397N were separately cloned into vectors to co-express fluorescent tags (mCherry or EGFP). These vectors were subsequently co-transfected into iPSCs. The double positive single cell was sorted by fluorescence activated cell sorting (FACS), and then cultured for further analysis (Fig. 2e, Supplementary Fig. 3c). A total of 25 single-cell clones were harvested and subjected to deep sequencing. As a result, these clones harbored G4300A conversion with frequencies that ranged from 1.14% to 72.97% (Fig. 2f and Supplementary Fig. 3d) and without integration of DdCBE pair in the genome (Supplementary Fig. 4a). We divided them into 4 groups according to the

editing efficiencies: 0-5%, 5-30%, 30-60% and over 60%. With the on-target editing efficiency increasing, the bystander editing at G4298 increased from 0.053 ± 0.003% to 2.160 ± 0.215% (Fig. 2f), which were well below the pathogenic threshold. We noticed that there is a significant correlation in editing efficiency between on-target and bystander sites in these clones (Supplementary Fig. 4b). Then we asked whether DdCBE-mediated G4300A correction was stable over time. Three clones (#54, #81 and #84) with over 70% of correction were cultured for 20 passages, and the editing efficiency of each clone changed from 71.90% to 80.58%, from 74.23% to 60.44%, and from 72.97% to 77.57%, respectively (Fig. 2g and Supplementary Fig. 4c). Accordingly, the bystander editing at G4298 changed from 1.96% to 2.10%, from 2.47% to 1.70%, and from 2.52% to 2.43% (Fig. 2h). Although the allele frequency of G4300A decreased slightly after 20 times of passage in one of the three clones, it remained high enough to achieve potential gene therapy. Meanwhile, the ratio of bystander mutation still kept at a very low level over time in these clones. Importantly, the edited iPSC clones still had normal karyotypes as expected and expressed pluripotent markers (Supplementary Fig. 4d–f). Together, the screened DdCBE pair can effectively correct the m.A4300G mutation in patient-derived iPSCs with limited bystander mutation.

## Off-target editing by DdCBE in mitochondrial and nuclear DNA.
To assess the off-target activity of DdCBE in the iPSC mitochondrial genome, we identified DdCBE induced SNVs by analyzing the whole mitochondrial genome sequencing data of untreated m.A4300G-iPSCs and 25 edited iPSC clones. Untreated m.A4300G-iPSCs showed the presence of naturally occurring SNVs (Supplementary Table 1). After excluding these

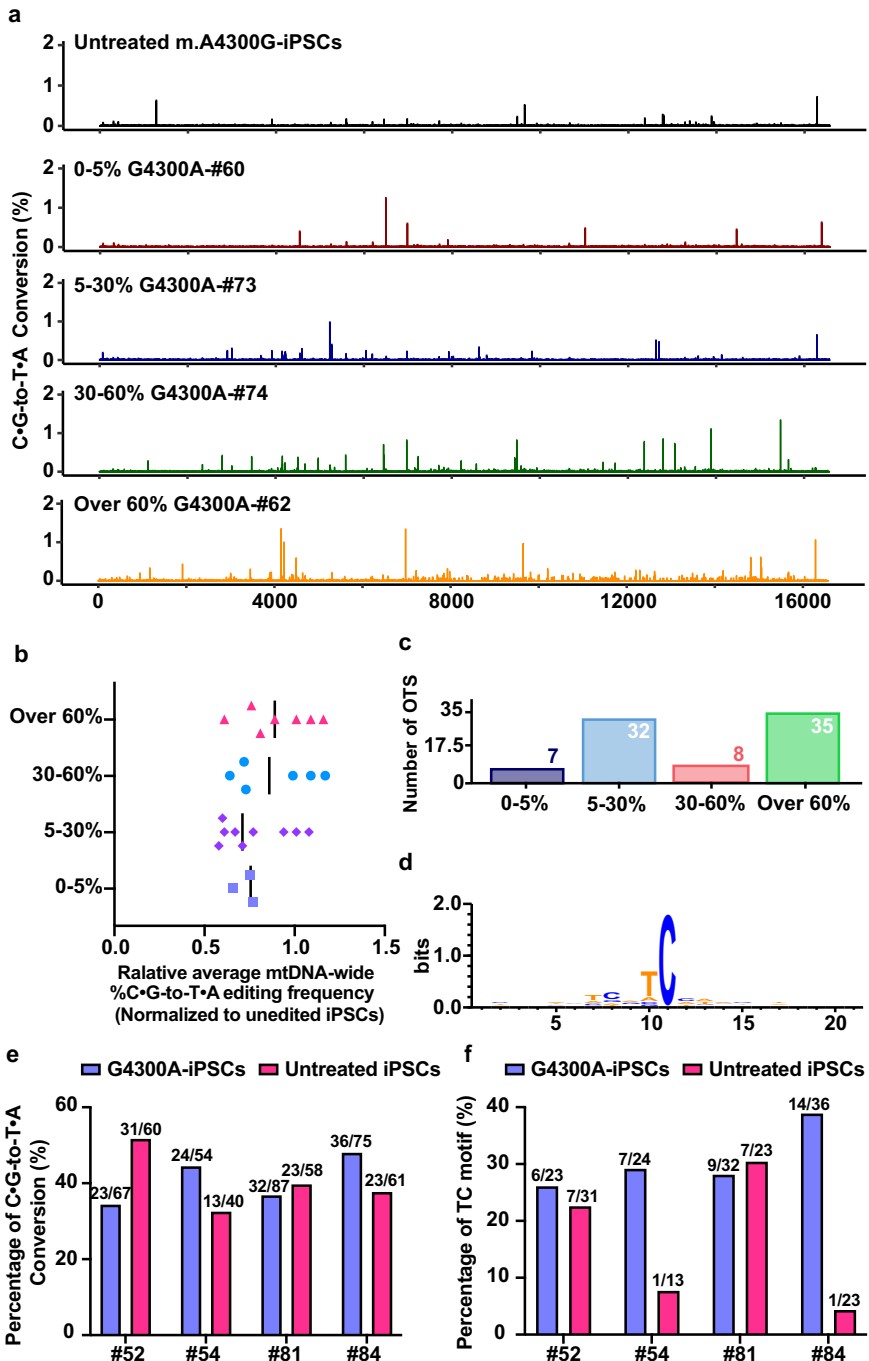

**Fig. 3 Off-target analysis of DdCBE on mtDNA and nDNA in corrected iPSC clones. a** Frequencies of C•G-to-T•A conversions along the whole mtDNA of corrected iPSC clones. Untreated m.A4300G-iPSCs were used as control. **b** Average frequency of mtDNA-wide C•G-to-T•A conversions for corrected iPSC clones in four groups with 0-5%, 5-30%, 30-60% and over 60% editing efficiencies, respectively. **c** The number of off-target sites in four groups of corrected iPSC clones. d. Sequence logos of the region flanking off-target sites. Bits reflect sequence conservation at a given position. **e** SNVs identified from nuclear genome of edited clones and untreated iPSCs. The number of C•G-to-T•A SNVs / Total number of identified SNVs are indicated on the top of each column. **f** Percentage of identified SNVs with tC motif in edited clones and untreated iPSCs. The number of C•G-to-T•A SNVs with tC motif / Total number of C•G-to-T•A SNVs are indicated on the top of each column.

SNVs, C·G-to-T·A conversions with frequency ≥1% in the 25 edited iPSC clones were selected for further off-target analysis (Supplementary Data 1). As shown in Fig. 3a and Supplementary Fig. 5, the efficiencies of DdCBE-mediated off-target ranged from 1.19% to 4.16% (group 0-5%), from 1.01% to 9.52% (group 5-30%), from 1.11% to 3.48% (group 30-60%) and from 1.05% to 9.10% (group over 60%). Strikingly, we could obtain the genetically corrected iPSC clones with off-target frequencies lower than

1.5% regardless of the on-target editing efficiency (Fig. 3a). In addition, the average frequencies of mitochondrial genome-wide C·G-to-T·A off-target editing in the edited iPSCs showed no significant difference among the four groups (Fig. 3b). To characterize the off-target sites, the distribution and motif of these sites were analyzed. The results revealed that the number of off-target sites showed no correlation with editing efficiencies (Fig. 3c). Among these mutations, most of them were detected

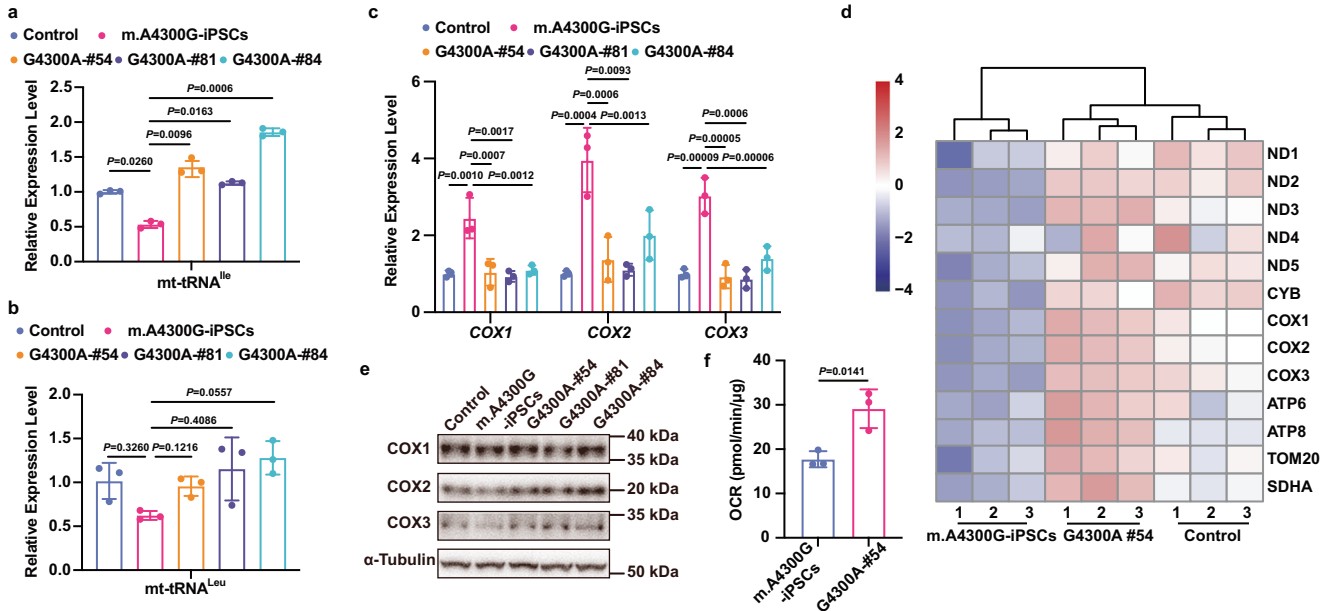

**Fig. 4 Correction of A4300G mutation restores the tRNA$^{Ile}$ level and mitochondrial function.** a and b. Verification of mt-tRNA$^{Ile}$(**a**) and mt-tRNA$^{Leu}$ (**b**) expression in control iPSCs, m.A4300G-iPSCs and corrected iPSC clones with editing efficiencies over 60% using qRT-PCR. Effect size for Control vs. m.A4300G-iPSCs, m.A4300G-iPSCs vs. G4300A-#54, m.A4300G-iPSCs vs. G4300A-#81 and m.A4300G-iPSCs vs. G4300A-#84 is 0.4667, −0.8223, −0.5933 and −1.323 in **a**; 0.3933, −0.3333, −0.5300 and −0.6567 in **b**. Data are presented as mean ± SEM, $n = 3$ independent experiments. Significance was calculated with One-way ANOVA. **c** Determination of *COX1*, *COX2* and *COX3* RNA expression in control, m.A4300G-iPSCs and corrected iPSC clones using qRT-PCR. Effect size for Control vs. m.A4300G-iPSCs, m.A4300G-iPSCs vs. G4300A-#54, m.A4300G-iPSCs vs. G4300A-#81 and m.A4300G-iPSCs vs. G4300A-#84 is −1.447, 1.410, 1.520 and 1.353 in *COX1*; −2.957, 2.587, 2.857 and 1.953 in *COX2*; −2.027, 2.103, 2.163 and 1.627 in *COX3*. Data are presented as mean ± SEM, $n = 3$ independent experiments. Significance was calculated with One-way ANOVA. **d** Targeted quantification of mitochondrial proteins using PRM. Each sample were conducted with three technical replicates. **e** Image of Western blot showed the protein level of COX1, COX2 and COX3 in control, m.A4300G-iPSCs and corrected iPSC clones. **f** The quantitative analysis of basal oxygen consumption rate (OCR) in m.A4300G-iPSCs and corrected iPSC clone by Seahorse assay (Effect size = −11.37). Data are presented as mean ± SD, $n = 3$ independent experiments. Significance was calculated with unpaired two-tailed Student's $t$-test.

within 5'-tC context, indicating the off-target editing was mostly caused by DddA$_{tox}$ (Fig. 3d). For off-target sites with editing efficiency over 5%, we found that these off-target editing events are largely independent of TALE-DNA interactions (Supplementary Fig. 6a).

To evaluate the potential influence of DdCBE activity on nuclear DNA, we performed whole genome sequencing for these edited clones including #52 from 0-5% group, and #54, #81 and #84 from over 60% group. We used Lofreq, Mutect2, and Strelka to call SNVs. The variants were identified in the edited clones with untreated m.A4300G-iPSCs as control, meanwhile, we also called variants in untreated m.A4300G-iPSCs with each edited clone as control. To reduce false positives, only SNVs identified by all the three algorithms were considered as the actual variants. Compared with the untreated m.A4300G-iPSCs, the SNVs with C·G-to-T·A conversions were not enriched in all the edited clones (Fig. 3e, Supplementary Data 2), but showed a preference for the tC context in #54 and #84 clones (Fig. 3f). Although 7 and 14 C·G-to-T·A SNVs with a preference for the tC context were detected in the two clones respectively, they are all not located within coding sequences (Supplementary Fig. 6b). These results suggest that no substantial nuclear off-target editing induced by DdCBE was detected in the edited iPSCs.

Taken together, DdCBE can achieve precise correction of A4300G mutation in the selected iPSC clones with limited nuclear and mitochondrial off-target editing.

**Correction of m.A4300G mutation restoring the tRNA$^{Ile}$ level and mitochondrial function.** To assess the effect of DdCBE treatment on the stability of mitochondrial tRNA$^{Ile}$ in the iPSCs,

we detected the expression of tRNA$^{Ile}$ using qRT-PCR in iPSCs with over 60% correction. The results revealed a significant increase of mitochondrial tRNA$^{Ile}$ in edited iPSCs (Fig. 4a), while the level of mitochondrial tRNA$^{Leu}$ kept relatively stable (Fig. 4b). We speculated that the restored mitochondrial tRNA$^{Ile}$ expression level may rescue the mtDNA-encoded protein level and mitochondrial function. qRT-PCR revealed the restored mRNA level of mitochondrial coding genes in corrected clones (Fig. 4c and Supplementary Fig. 6c). To probe the levels of mtDNA-encoded proteins, we successfully designed unique peptides to target 11 proteins, except ND4L and ND6 (Supplementary Table 2). Parallel reaction monitoring (PRM) results clearly identified the lower protein levels of 11 mitochondrial genes in patient-derived iPSCs than that in control iPSCs, while in the corrected clone, the protein levels were restored to the control levels (Fig. 4d and Supplementary Fig. 7). To further verify the restored protein levels, G4300A-#54, #81 and #84 with editing efficiencies over 60% were used for western blot. In agreement with the PRM results, the protein levels of COX1, COX2 and COX3 were restored (Fig. 4e and Supplementary Fig. 6d). In addition, compared with m.A4300G-iPSCs, the corrected clone exhibited a restored basal rate of oxidative phosphorylation (Fig. 4f). Taken together, these results demonstrate that DdCBE-mediated correction of m.A4300G mutation results in the recovery of tRNA$^{Ile}$ steady-state levels, the restored protein levels of mitochondrial genes, and the rescue of mitochondrial function.

## Discussion

Due to the substantial role of pathogenic mtDNA mutation in the occurrence of PMDs, effective mtDNA editing technology has been increasingly demanded from the clinical side. Although

mitochondrially targeted restriction enzymes and programmable nucleases were the first successful strategies to achieve mtDNA genetic manipulation through selective cleavage of mutant mtDNA and reduction of mutation load[9,11,28], DdCBE is a more direct and effective strategy to convert the mutant mtDNA, and is more broadly applicable for correction of both heteroplasmic and homoplasmic mtDNA mutations[16–18,20]. Our study demonstrates the potential of DdCBE-based therapy for treating PMDs by using the patient derived iPSC carrying a pathogenic homoplasmic m.A4300G mutation. With the continuing improvement in the DdCBE technology, it can be expected that more pathogenic mtDNA mutations would be correctable with increasing efficiency and reduced off-target effects[29,30].

Although DdCBE can mediate C·G-to-T·A conversion at designed loci precisely, it is also known to cause noticeable off-target editing in both mitochondrial and nuclear genome[31,32]. Previous studies showed that the degree of mitochondrial off-target editing caused by DdCBE varied with specific DdCBE pairs in different cell lines and animal models[16–18,20]. In this study, our data indicated that genetically corrected iPSC clones could be obtained with off-target frequencies lower than 1.5% regardless of the on-target editing efficiency. Recently, individual Mitochondrial Genome sequencing (iMiGseq) was developed to achieve ultra-sensitive variant detection and unbiased evaluation of heteroplasmy levels at individual mtDNA molecule level and single cell level[33,34]. Thus, iMiGseq would help address two unexplored aspects: accurately measuring heteroplasmy levels by counting molecules directly in each genetically corrected iPSC clone, and examining the connection between off-target editing and on-target editing within single mtDNA molecules. As for the nuclear off-target editing caused by DdCBE, our study showed that only a few nuclear off-target sites were detected in certain edited iPSCs. Therefore, at present, the specificity of DdCBE is adequate to generate edited iPSC clones with limited off-target editing, as long as sufficient clones are tested.

Among current genetical treatments of PMDs, mitochondrial replacement therapy (MRT) has the ability to prevent the inheritance of pathogenic mtDNA mutations, but this therapy remains ethically controversial and has yet to be approved in most countries[35]. Recently, several preclinical studies of adeno-associated virus (AAV)-vector based gene therapies showed promising for PMDs[36]. Up to now, an AAV2 vector-based gene therapy product for the treatment of LHON, delivering the *MT-ND4* gene in the mitochondria, has been under-evaluated in clinical trial[37]. Herein, we demonstrated that DdCBE is able to effectively and specifically repair the homoplasmic m.A4300G mutation in patient-derived iPSCs.

In terms of the development of related therapies, it has been demonstrated that DdCBE can install mutations on mtDNA by delivering DdCBEs into wild-type mouse heart using AAV vectors[38], suggesting the possible feasibility to directly edit the patient's cardiomyocytes. However, mitochondrial diseases are extremely complicated with clinically heterogeneous. It is still unknown whether damaged mitochondrial function can be restored through genetic correction and what proportion of mtDNA mutations corrected can rescue the phenotype. More importantly, preclinical studies using patient-derived iPSCs are necessary before DdCBE can be used for the clinical treatment of mitochondrial diseases. In this study, we focused on human pathogenic mtDNA mutations and conducted a proof of concept study to demonstrate that a screened DdCBE could mediate base editing to correct the homoplasmic m.A4300G mutation, thus restoring impaired mitochondrial function in patient-derived iPSCs. It is important to note that our work represents a small preliminary step in this field, and substantial further research is still required to advance the early clinical application of this technology for treating patients with mitochondrial diseases.

We propose several strategies to further optimize DdCBE for mitochondrial gene therapy: (1) Use evolved DddAs and DddA homologs to enhance DdCBE's activity and expand its targeting scope on mtDNA[39–41]; (2) Fuse transactivators (VP63, P65 or Rta) to the end of UGI to improve the frequencies of C-to-T conversions[40]; (3) Substitute DddA$_{tox}$ with high-fidelity DddA to reduce mitochondrial off-targets[30]; (4) Introduce nuclear export signal (NES) to the DdCBE system or simultaneously express DddI$_A$ in the nucleus to reduce nuclear off-targets[32]; (5) DNA strand-selective mitochondrial base editors could be an alternative strategy to reduce bystander editing[42]. These strategies may help obtain genetically corrected iPSC clones more efficiently and precisely. Additionally, our study suggests considering several factors when utilizing DdCBE, including the number of tC motif within the spacing region, the length of the spacing region, the contexts of TALE binding sites, and the orientation and split types of DddA$_{tox}$. All of these factors collectively contribute to the performance of DdCBE pairs.

Consistent with the phenotypic and genetic heterogeneity of PMDs, the pathogenesis of the mtDNA mutations is difficult to be evaluated and confirmed, largely due to the lack of proper tools for functional analysis on certain mtDNA variants. In this study, we found the patient's mother carrying the same homoplasmic m.A4300G mutation with no related phenotype, which is similar to previous studies on m.A4300G[22]. Our data showed that the phenotypes including mitochondrial protein level and mitochondrial respiration were restored by repairing m.A4300G mutation back to wild-type allele at over 60%, confirming that the impaired mitochondrial function is truly caused by this mutation. Our results provide a reference for developing any gene-editing based therapy strategies for targeting this mutation site in the future.

In summary, here we corrected the homoplasmic m.A4300G mutation via DdCBE in patient-derived iPSCs, enabling the recovery of tRNA$^{Ile}$ steady-state levels, protein levels of mitochondrial genes, and the mitochondrial functions correspondingly. We envisage that our strategy reported here would be used for more types of PMDs with the optimization of DdCBE.

## Methods

**Human induced pluripotent stem cells (iPSCs) Reprogramming and Culture.** This study was ethically approved by the Medical Ethics Committee of Nanjing Maternal and Child Health Care Hospital (2021KY-131), and informed consents were obtained from the patient's legal guardian as well as the healthy donors, in accordance with the Declaration of Helsinki. All ethical regulations relevant to human research participants were followed.

PBMC from both a HCM patient with homoplasmic m.A4300G mutation and a healthy donor were reprogrammed using Sendai viral vectors (Thermo Fisher Scientific, A16517). Briefly, a total of $5 \times 10^5$ PBMCs were plated and cultured in complete PBMC medium (StemPro-34 Medium supplemented with 100 ng/mL SCF, 100 ng/mL FLT-3, 20 ng/mL IL-3 and 20 ng/mL IL-6 (ThermoFisher Scientific) for 4 days, and transduced using CytoTune 2.0 Sendai reprogramming vectors at the appropriate MOI (KOS MOI = 6, hc-Myc MOI = 6, hKlf4 MOI = 4) overnight. After transduction, cells were harvested and seeded on plates coated with Geltrex Matrix (Thermo Fisher Scientific) in complete StemPro-34 medium without cytokines for 3 days. The medium was then changed to StemFlex medium (Thermo Fisher Scientific) and replaced every day. On days 15–28 after transduction, iPSC clones could be seen and single clone was manually picked and

passaged. The iPSC clones used in this study were tested negative for mycoplasma.

**Cardiomyocyte differentiation of iPSC.** STEMdiff™ Cardiomyocyte Differentiation Kit (Stem Cell Technologies) was used to perform the cardiomyocyte differentiation according to the manufacturer's protocol. Briefly, iPSCs were seeded on 12-well plates coated with Matrigel at a density of $3.5 \sim 8 \times 10^5$ cells/well and cells must reach >95% confluency before starting differentiation. On day 0 of differentiation, medium was changed to 2 mL Medium A supplemented with Matrigel (1 in 100 dilution). On day 2, full-medium change was performed with Medium B and on day 4 and 6 with Medium C. From day 8, cells were maintained with STEMdiff™ Cardiomyocyte Maintenance Medium with full-medium change every 2 days for a month or longer. On day 8, small areas of beating cardiomyocytes could be seen, and on day 10, larger areas of beating cardiomyocytes could be seen over time. Cardiomyocytes were dissociated with STEMdiff™ Cardiomyocyte Dissociation Kit (Stem Cell Technologies) for following assays.

**Fluorescent Immunocytochemistry.** Cells were washed by PBS and then fixed by 4% paraformaldehyde (PFA) for 15 minutes at room temperature. After washing cells with PBS twice, cells were permeabilized by PBS containing 0.1% Triton X-100 (PBST, Sigma) for 20 minutes. Then cells were blocked in 1% bovine serum albumin (BSA, Sigma) in PBST for half an hour at room temperature. After that, cells were incubated with the primary antibody (Invitrogen) 1:100 diluted in 1% BSA in PBST overnight at 4 °C. Next day, after being washed, cells were incubated with fluorescence-labelled secondary antibodies (Invitrogen) 1:1000 diluted in PBS for at least 1 hour at room temperature while protected from light. Finally, the nuclei were counterstained with DAPI (Invitrogen) for 1 minute at room temperature. Results were imaged by the DMi8 fluorescent microscope (Leica).

**Karyotyping.** For cytogenetic analysis of the iPSC lines generated from the m.A4300G patient, GTG banding at the 400 to 550-band level was performed according to a standard protocol. Briefly, cells were treated with colchicine and 0.8% hypotonic disodium citrate at 80% ~ 90% confluency, followed by fixation (3:1 ratio of methanol and acetic acid). After dropped on pre-cooled glass slides and dried on 56 °C overnight, the cells were stained with Giemsa for 5 minutes. Twenty metaphases were counted and karyograms were analyzed using cytovision software.

**Western blot.** Cells were collected and homogenized by lysis buffer (50 mM Tris-Cl pH 7.4, 150 mM NaCl, 1% Triton X-100, 0.1% SDS and protease inhibitors cocktail). 30 μg protein of each sample was loaded onto 11% gels for electrophoresis, and then transferred to a PVDF membrane (Millipore). Membranes were incubated with primary antibody overnight at 4 °C using anti-COX1 (A17889, ABclonal; 1:1000), anti-COX2 (55070-1-AP, Proteintech; 1:1000), anti-COX3 (A9939, ABclonal; 1:1000) and anti-α-Tubulin (AF0001, Beyotime Biotechnology; 1:1000), followed by wash with TBST and incubation with Horseradish peroxidase (HRP)-linked second antibody (BL001A, Biosharp; 1:5000) for 1 hour at room temperature. Signal was detected with enhanced chemiluminescence detection reagent (Vazyme) and imaged by Tannon4500 SF.

**Plasmid construction.** All DdCBE vectors were assembled using our DdCBE assembly kit (Addgene, Kit #1000000212)[18]. DdCBE expression backbone and RVDs plasmids were digested with Bsa I (New England Biolabs) and ligated with T4 DNA ligase (New England Biolabs) in a single tube using the following program:

37 °C for 10 min; 10 cycles of 10 min at 37 °C and 10 min at 16 °C; 50 °C for 5 min; 80 °C for 5 min. The assembled plasmids were chemically transformed into *Escherichia coli* DH5α competent cells (Transgene), and then confirmed by PCR and Sanger sequencing. Through Pme I (NEB) and Not I (NEB) double enzymes digestion, DdCBE pair with best performance was cloned into DdCBE expression backbone with EGFP or mCherry tag.

For construction of pEGFP-N1 plasmid containing the m.A4300G target sequence, the synthesized sense and antisense oligos containing the target sequence were anneled and amplified by Green Taq Mix (Vazyme) for 3 cycle to leave A tail, and then ligated to Xcm I (NEB) digested pEGFP-N1 plasmid with T4 DNA ligases (NEB). Sequences of primers and DdCBE plasmids are listed in Supplementary Table 3 and Supplementary Note.

**Plasmid nucleofection.** $2 \times 10^5$ HEK293FT cells were nucleofected using the SE Cell Line 4D-Nucleofector X Kit (Lonza, program DS-150) according to the manufacturer's protocol. 500 ng of left DdCBE monomer and right DdCBE monomer each with 50 ng pEGFP-N1 containing A4300G target site were used for each nucleofection. The nucleofected cells were treated with 2 μg/mL puromycin 24 h post nucleofection, and collected at day 3 to recover the target pEGFP-N1 palsmids and amplify them for Sanger sequencing and/or deep sequencing. This cell line was tested negative for mycoplasma. $2 \times 10^5$ iPSCs were nucleofected using the P3 Primary Cell 4D-Nucleofector X Kit (Lonza, program CA-137) according to the manufacturer's protocol. 800 ng of left EGFP-DdCBE monomer and right mCherry-DdCBE monomer each were used for each nucleofection. 48 h post-nucleofection, the cells were dissociated into single cell, and EGFP and mCherry double positive single-cells were sorted into a 96-well plate. The representative gating strategy for EGFP/mCherry double positive cells is shown in Supplementary Fig. 8. The single-cell clones were collected for further analysis.

**qRT-PCR.** Total RNA was extracted with TRIzol reagent (Invitrogen) according to the manufacturer's protocol. After removing the residual genomic DNA, 1 μg of total RNA was reverse-transcribed into cDNA using the HiScript III RT SuperMix for qPCR (Vazyme). Then qRT-PCR was performed with ChamQ Universal SYBR qPCR Master Mix kit (Vazyme) using an Applied Biosystems 7500 Real-Time PCR System (Applied Biosystems, Forster City, CA).

To determine the tRNA expression level, the stem-loop primers for tRNA were used for reverse transcription following the manufacturer's protocol (Vazyme, and qRT-PCR analysis was performed as above mentioned. The relative expression of tRNA was normalized to U6 RNA and then calculated using the comparative Ct method ($2^{-\Delta\Delta CT}$). The primer information is listed in Supplementary Table 3.

**In-vitro trilineage differentiation of iPSC.** In vitro three germ lineages differentiation was performed with STEMdiff™ Trilineage Differentiation Kit (Stem Cell Technologies) according to the manufacturer's protocol. On day 0, $8 \times 10^6$ iPSCs were seeded on 12-well plates for differentiation to the ectoderm and endoderm, and $2 \times 10^6$ iPSCs for differentiation to the mesoderm. From day 1, cells were fed daily with 1.5 mL/well lineage-specific medium and then harvested on day 5 for mesoderm lineages and endoderm lineages and day 7 for the ectoderm lineages for following assays.

**tRNA-seq and data analysis.** Transfer RNA sequencing was conducted as YAMAT-seq reported previously[43]. In brief, 1 μg of total RNAs were incubated at 37 °C for 40 min in 20 mM Tris-

HCl (pH 9.0) to remove amino acids from mature tRNAs. Deacylated tRNAs were treated with T4 polynucleotide kinase (NEB) at 37 °C for 30 min to further warrant a free 3' hydroxyl group. The annealed Y-shaped adaptors were ligated to pretreated tRNAs using T4 RNA ligase 2 according to the manufacturer's protocol (NEB). Ligated tRNAs were incubated with reverse transcription primer at 70 °C for 2 min and then placed on ice immediately. Reverse transcription was subsequently performed using SuperScript III Reverse Transcriptase kit according to the manufacturer's protocol (Thermo). The resultant cDNAs were amplified for 11 cycles using KAPA HiFi HotStart ReadyMix (Roche). Libraries were gel extracted and pooled for sequencing by Illumina NovaSeq platform.

The tRNA-seq data analysis was conducted as previously described[43]. The R1 reads of PE150 were chose for further analysis. Reads were quality control (QC) by FastQC (v0.11.5) and adapters were trimmed by cutadapt (v1.15). A set of 632 tRNA-reference genes (listed in gtRNAdb) was used for reference and CCA nucleotide sequence was added to the tail if needed. Reads mapping was performed by using Bowtie2 (v2.3.4.1) and alignments with high quality were kept for further analysis. Differential analysis was performed using R package DESeq2 (v1.32.0) and ggplot2 (v2.0.0).

**Relative protein quantification by parallel reaction monitoring (PRM)**. The samples were lysed by lysis buffer consisting of 8 M urea and protease/phosphatase inhibitors in Tris-HCl (pH 8.0), then sonicated for complete lysis. The extracted proteins from samples were reduced and alkylated by DTT and IAA, respectively. Protein digestion was performed using trypsin (5 ng/μL) for 16 h at 37 °C. Finally, the peptides were desalted through the C18 columns and then dried before use.

All the crude isotope-labeled heavy synthetic peptides (Supplementary Table 2) were all purchased from Synpeptide. Each peptide sample (500 ng) combined with the isotope-labeled peptides was separated by an analytical column (75 μm, 1.7 μm × 15 cm, USA) using a flow rate of 300 nL/min on an easy-nLC 1200 HPLC system (Thermo Fisher Scientific) and a 60-minute gradient. Analysis was conducted using a scheduled method on the Orbitrap Fusion™ Lumos™ mass spectrometer with the following parameters: a higher-energy collision of 30 eV, an AGC target of 5E4, a maximal injection time of 118 ms, and a scan range (m/z) of 150–2,000. Three technical replicates were performed for each group.

PRM data were processed with Skyline Daily software. At least three transitions per precursor were used to quantify the targeted peptides in the samples. For relative quantification, the protein expression was calculated as the ratio of the endogenous to the heavy peptide, which was then added to the transition peak areas. Isotope-labeled heavy peptides were spiked into tryptic peptides of bovine serum albumin, serving as negative control. To exclude the batch effect among the three groups, we used a housekeeping protein β-ACTIN to perform normalization.

**Deep sequencing and data analysis**. The first round PCR (PCR1) was conducted to specifically amplify target region containing A4300G site using Phanta Max Super-Fidelity DNA Polymerase (Vazyme). Then gel extracted PCR1 products were used for the second round PCR (PCR2) with barcoded primers. The products of PCR2 were pooled with equal moles and further amplified using index primers (Vazyme) and purified by 1.0 × DNA Clean beads for sequencing on the Illumina NovaSeq platform. Primers used in PCR1 and PCR2 are listed in Supplementary Table 3. The deep sequencing data analysis were conducted as described in a previous report[20]. Briefly, the reference sequence of human

mitochondrial genome (NC_012920.1) from the NCBI database was used for alignment by bowtie2. Alignment results were converted to bam format using samtools and visualized in Integrative Genomics Viewer (IGV). Bases with depth over 2 million were truncated to 2 million, and only C-to-T or G-to-A conversion was calculated for DdCBE-mediated editing.

**Whole mtDNA sequencing and data analysis**. Whole mtDNA was captured by long-range PCR[20]. Two overlapping mtDNA fragments around 8 kb each were purified by gel extraction and subjected to library preparation using TruePrep DNA Library Prep Kit V2 for Illumina (Vazyme). Libraries were pooled and sequenced by Illumina NovaSeq platform. The quality control of raw data was performed by fastqc and trim_galore in paired end mode. After trimming the Illumina adapter sequence or Ns in either side of the read, only reads with quality over 20 were mapped to NC_012920.1 by using bowtie2 with default parameters of paired end. The DdCBE editing efficiency was calculated as mentioned above.

**Whole genome sequencing and data analysis**. Genomic DNA extracted from iPSCs was subjected to library preparation using the KAPA HyperPlus Kit (KR1145) according to the manufacturer's instruction. Whole genome sequencing was performed at mean coverage of ~10 × using Illumina NovaSeq platform. The data analysis was performed as described in a previous study[31] with minor modifications. Briefly, the trimmed reads were mapped to the human reference genome (GRCh38/hg38) by BWA (v0.7.12). Picard-tools (v2.3.0) was used to reorder, sort, add read groups and mark duplicates of the aligned BAM files. Then, Strelka (v2.9.10), Lofreq (v2.1.2) and Mutect2 (v3.8.1) were used to identify the genome-wide de novo variants with high confidence. The variants were identified in the mapped BAM file of gene correction iPSC with m.A4300G-iPSC as control. Also, we called variants in m.A4300G-iPSC with gene correction iPSC as control. Only SNVs identified by all the three algorithms were used for the further analysis. Variants were removed if they overlapped with (1) repeat regions and microsatellite sequences downloaded from UCSC genome browser (http://genome.ucsc.edu/), and (2) genetic variant sites reported in dbSNP151 database (https://www.ncbi.nlm.nih.gov/snp/).

**Off-target analysis**. For the off-target analysis, the following sites were excluded before analysis and visualization: (1) SNP sites with C·G to T·A variation of East Asian were obtained from the NCBI database; (2) the sites of which C·G to T·A variation over 1% in untreated A4300G iPSCs; (3) sites within the DdCBE spacing region. Excluded SNPs are listed in Supplementary Table 1. The average off-target editing frequency for each sample was calculated as: (summed reads of C·G to T·A conversion) / (total number of reads that covered all non-target C·G base pair).

**Seahorse mito-stress test**. The Seahorse XFe24 Analyzer (Agilent) was used to assess metabolic ability of iPSCs according to the manufacturer's protocol. Cells were seeded on XFe24 plates coated with Matrigel at a density of $3 \times 10^4$ overnight and maintained with corresponding medium. One hour before the test, medium was changed to the Seahorse XF base medium supplemented with 25 mM Glucose, 1 mM Pyruvate and 2 mM L-Glutamine and then cells were cultured in a 37 °C, $CO_2$-free incubator for one hour. The XF Cell Mito Stress Test Kit was used to perform the test with the following concentrations of injected compounds: 0.5 μM oligomycin, 2.5 μM FCCP (Carbonyl cyanide 4- (trifluoromethoxy) phenylhydrazone), 1.5 μM rotenone and antimycin A.

**Statistics and Reproducibility.** All experiments were repeated at least three times. Statistical analyses were performed with Student's t test or One-way ANOVA using SPSS version 23.0 statistical software (IBM). The data are presented as the mean ± SEM or mean ± SD as indicated in the figure legend, and the differences between groups were considered statistically significant when $P$ value < 0.05.

**Reporting summary.** Further information on research design is available in the Nature Portfolio Reporting Summary linked to this article.

## Data availability

The high-throughput sequencing data have been deposited to the NCBI Sequence Read Archive (SRA) database under the accession ID PRJNA921944. The source data behind the graphs and charts in the main figures are included in Supplementary Fig. 9 and Supplementary Data 3. Plasmids generated in this study have been deposited to Addgene (210395, 210396). All other data are available from the corresponding authors upon reasonable request.

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

## Acknowledgements

This work was supported by the National Key R&D Program of China (2021YFC2700600, 2022YFC2702705), the Funds for Creative Research Groups of China (82221005), the National Natural Science Foundation of China (31970796), the China Postdoctoral Science foundation (2022T150332) and the Basic Research Program of Jiangsu Province (BK20220315), the Medical Science and technology development Foundation of Jiangsu Commission of Health (ZD2021058).

## Author contributions

B.S. and D.L. conceived the project and designed the experiments. Z.X., S.Y and L.J performed clinical analysis. X.C., M.C., Y.Z. and J.Z. performed plasmids construction, sequencing and cell culture. Y.W. and X.G. performed proteomics and bioinformatics analysis. H.S. and C.W. performed all bioinformatics analyses with the help of Y.X. and Z.H. B.S., D.L. and X.C. wrote the manuscript with inputs from all authors.

## Competing interests

The authors declare no competing interests.
