## [Peer Review File · Communications Biology]

Reviewers' comments:

Reviewer #1 (Remarks to the Author):

In this study, the authors established and characterized the patient-derived iPSC carrying the homoplasmic mitochondrial A4300G mutation (m.A4300G-iPSC). They performed the screening assay in HEK293FT cells and generated the best m.A4300G targeting DdCBE pairs. Using the best DdCBE pairs, the authors efficiently corrected the m.A4300G mitochondrial mutation in the patient-derived iPSC with minimal bystander mutation. Whole mitochondrial genome sequencing of the 25 edited iPSC clones revealed a moderate off-target mutation in mitochondrial DNA (mtDNA). Whole-genome sequencing also revealed minimal off-target effects in nuclear DNA in the edited iPSCs. In addition, the authors showed that the corrected clones restored mitochondrial tRNA^{Ile} expression levels, protein levels, and mitochondrial function.

Overall, the experiments are well done and support the conclusions. Please clarify the following points before publication

1. In Figure 1f, the expression level of mt-tRNA^{Ala} was increased in the m.A4300G-cardiomyocytes. Is there any reason that can explain this? Please describe it in the main text.
2. The screening system in HEK293FT is not clear. How was the pEGFP-N1 plasmid containing the m.A4300G target sequence constructed? Why did the author use NLS-DdCBD? If the authors target nuclear DNA, how can they isolate the nuclear-localized pEGFP-N1 plasmid from the cytoplasm-localized one? This information is missing in the current manuscript. The authors should describe in more detail about the screening system used to generate m.A4300G targeting DdCBEs.
3. In Figure 2f, is there a correlation in editing efficiency between on-target and bystander sites each clone?

Reviewer #2 (Remarks to the Author):

The authors generated iPSCs with the homoplasmic m.A4300G mutation from an HCM patient. The function of mitochondria in this cell model is compromised. After correction by DdCBE pair, the corrected clone exhibited the rescue of mitochondrial function. As for proof-of-concept, Michal Minczuk has demonstrated the feasibility using of DdCBEs in vivo in cell model and somatic tissue (In vivo mitochondrial base editing via adeno-associated viral delivery to mouse post-mitotic tissue, Nature Communication, 2022). The similarity between correcting homoplasmic mutations and establishing disease models is that their starting cells are both homoplasmic. In short, this study obtained edited iPSC clones by applying DdCBE for the first time. There are some small issues.

1. The authors use plasmid to deliver DdCBE into cells, so the risk of RNA off-target in iPSC induced by DdCBE (including deaminase) should be evaluated. Random integration also should be tested.
2. Line 41, the author applied DdCBE with different TALE pair to target mitochondrial DNA, technically, it is not optimized, it is screened or selected. "An optimized DdCBE pair" seems like "the whole DdCBE system has been upgraded".
3. Cells after gene editing need to be tested for pluripotency and karyotype.
4. In Fig. 4e, COX1 and COX2 protein levels seem different from Fig 1. Completing quantitative analysis would be better.
5. Flow cytometry figures about iPSCs transfection efficiency should be provided

Reviewer #3 (Remarks to the Author):

This manuscript presented a patient-derived induced pluripotent stem cell (iPSC) line generated from a pathological specimen, aiming to address the m.A4300G mutation through the application of Dd-CBE. The ensuing experiment effectively demonstrated the notable benefits of high gene editing efficiency and reduced bystander effects. Nevertheless, several questions remain.

1. Primarily based on the fundamentals of Dd-CBE, the gene editing strategy employed in this study raises the inquiry of whether any novel elements were introduced. Additionally, considering the relative frequency of Dd-CBE usage in iPSC research, it is crucial to ascertain any advancements compared to prior investigations.
2. Regarding the therapeutic strategy associated with this iPSC gene editing pipeline, an essential aspect to clarify is whether the objective is to utilize edited iPSCs directly or to derive cardiomyocytes from these edited iPSCs. Is it feasible to directly edit the patient's cardiomyocytes? Substantial evidence is still required to establish the viability of employing Dd-CBE in gene therapy.
3. Apart from gene editing efficiency and off-target effects, are there any other factors that necessitate consideration when utilizing Dd-CBE? Considering the distinct characteristics of each patient, should Dd-CBE pair screening still be conducted? Furthermore, what specific attributes contribute to the successful performance of Dd-CBE pairs?

MS number: COMMSBIO-23-1172-T

MS title: Correction of homoplasmic mitochondrial tRNA mutation in patient-derived iPSCs via mitochondrial base editor

Authors: Chen *et, al*

Response to reviewers' comments

We greatly appreciate the comments and suggestions from the reviewers, and the opportunity to improve the manuscript. We have carefully considered each of the comments and addressed them as detailed in the following response. The changes made to the manuscript are highlighted in yellow in the main text. We have outlined our responses to individual reviewer comments below. Each comment is shown in black, and is followed by our response in blue. We also include the revised text (italics and underlined) within the reply.

Comments from reviewer(s):

Reviewer #1 (Remarks to the Author):

In this study, the authors established and characterized the patient-derived iPSC carrying the homoplasmic mitochondrial A4300G mutation (m.A4300G-iPSC). They performed the screening assay in HEK293FT cells and generated the best m.A4300G targeting DdCBE pairs. Using the best DdCBE pairs, the authors efficiently corrected the m.A4300G mitochondrial mutation in the patient-derived iPSC with minimal bystander mutation. Whole mitochondrial genome sequencing of the 25 edited iPSC clones revealed a moderate off-target mutation in mitochondrial DNA (mtDNA). Whole-genome sequencing also revealed minimal off-target effects in nuclear DNA in the edited iPSCs. In addition, the authors showed that the corrected clones restored mitochondrial tRNA^{Ile} expression levels, protein levels, and mitochondrial function.

Overall, the experiments are well done and support the conclusions. Please clarify the following points before publication.

We thank the reviewer for the positive comments.

1. In Figure 1f, the expression level of mt-tRNA^{Ala} was increased in the m.A4300G-cardiomyocytes. Is there any reason that can explain this? Please describe it in the main text.

Thanks for the comment. We have also noticed this result, but really don't know the exact mechanism. We went through the previous literatures and found a possible explanation. An increased level of lactate in body fluids serves as a biochemical marker in assessing patients with mitochondrial disorders. Because alanine is formed from pyruvate by alanine transaminase at a rate proportional to intracellular lactate levels, increased alanine level is also often found in patients with mitochondrial disorders (PMID: 21075032, 7832568, 9054475, 16854608). Cardiomyocytes have stronger metabolic activity than iPSCs, and their mitochondrial dysfunction may cause more

pyruvate/lactate accumulation in m.A4300G-cardiomyocytes, ultimately leading to increased alanine level. We speculate that the elevated alanine level might regulate the expression level of its tRNA counterpart in a feedback manner. We have added the possible explanation to the main text at Line 147-153: Unexpectedly, we also observed the up-regulation of mitochondrial tRNAAla in m.A4300G-cardiomyocytes (Fig.1f). Increased alanine level is often found in patients with mitochondrial disorders, because mitochondrial dysfunction may cause more pyruvate accumulation, which can then be converted to alanine by alanine transaminase. We speculate that the up-regulation of mitochondrial tRNAAla is probably a response to the elevated alanine level in m.A4300G-cardiomyocytes.

2. The screening system in HEK293FT is not clear. How was the pEGFP-N1 plasmid containing the m.A4300G target sequence constructed? Why did the author use NLS-DdCBE? If the authors target nuclear DNA, how can they isolate the nuclear-localized pEGFP-N1 plasmid from the cytoplasm-localized one? This information is missing in the current manuscript. The authors should describe in more detail about the screening system used to generate m.A4300G targeting DdCBEs.

Thanks for the comments. We are sorry we did not describe the screening system clearly.

We have added the information on the construction of pEGFP-N1 plasmid containing the m.A4300G target sequence and more details about the screening system in the section of methods (Line 422-425, For construction of pEGFP-N1 plasmid containing the m.A4300G target sequence, the synthesized sense and antisense oligos containing the target sequence were annealed and amplified by Green Taq Mix (Vazyme) for 3 cycle to leave A tail, and then ligated to Xcm I (NEB) digested pEGFP-N1 plasmid with T4 DNA ligases (NEB).).

Due to the low transfection efficiencies in iPSCs, it is not easy to screen DdCBE pair with best performance to correct m.A4300G in mitochondria. To accelerate the screening process, we did it in HEK293FT cell line by co-nucleofection of NLS-

DdCBE pairs and a pEGFP-N1 plasmid harboring the m.A4300G target sequence. Previously, we also used the system to screen high-efficient DdCBEs for zebrafish, due to the lack of available cell lines for zebrafish (PMID: 34480028).

Nucleofection uses the combination of a specific nucleofector solution and specific electrical parameters to achieve the delivery of plasmid DNA directly into the cell's nucleus (PMID: 15121170). We believe that a small amount of plasmid remains in the cytoplasm because of the limitations of nucleofector technology, and the remaining cytoplasm-localized ones could be edited by NLS-DdCBE when nuclear envelope disappeared or by cytoplasm-retained NLS-DdCBE. Therefore, we did not isolate the nuclear-localized and cytoplasm-localized pEGFP-N1 plasmids, and amplified them both for Sanger sequencing and deep sequencing. We have added more details about the screening system to the main text. Please find the Line 169-172: *As previously reported, we screened each NLS-DdCBE pair by co-transfecting them with the pEGFP-N1 plasmid harboring the m.A4300G target sequence into HEK293FT cells. The target pEGFP-N1 plasmid was recovered and amplified for Sanger sequencing 72 hrs post-nucleofection.*

3. In Figure 2f, is there a correlation in editing efficiency between on-target and bystander sites each clone?

Thanks for the valuable suggestion. As shown in new Supplementary Fig.4b, there is a significant correlation in editing efficiency between on-target and bystander sites each clone. We have added the description in the main text, Line 197-199: *We noticed that there is a significant correlation in editing efficiency between on-target and bystander sites in these clones (Supplementary Fig.4b).*

new Supplementary Fig.4b

b. Correlations between on-target editing efficiencies and bystander editing efficiencies in edited clones.

Reviewer #2 (Remarks to the Author):

The authors generated iPSCs with the homoplasmic m.A4300G mutation from an HCM patient. The function of mitochondria in this cell model is compromised. After correction by DdCBE pair, the corrected clone exhibited the rescue of mitochondrial function. As for proof-of-concept, Michal Minczuk has demonstrated the feasibility using of DdCBEs in vivo in cell model and somatic tissue (In vivo mitochondrial base editing via adeno-associated viral delivery to mouse post-mitotic tissue, Nature Communication, 2022). The similarity between correcting homoplasmic mutations and establishing disease models is that their starting cells are both homoplasmic. In short, this study obtained edited iPSC clones by applying DdCBE for the first time. There are some small issues.

Thanks for the valuable suggestions. Previously, we and other groups successfully generated mitochondrial disease models by injection of DdCBE mRNAs into early embryos (PMID: 35102133, 34938607, 35102135, 34663794, 34480028, 33608520). In Michal Minczuk's work, they installed mtDNA mutations in adult and neonatal wild-type mice by delivering DdCBE into mouse heart using adeno-associated virus (AAV) vectors, suggesting the potential translation to human gene correction therapies. However, as we know, mitochondrial diseases are very complicated with clinically heterogeneous, no one knows whether damaged mitochondrial function can be rescued by genetic correction, and what proportion of mtDNA mutations are corrected can rescue the phenotype. More importantly, preclinical studies are required before DdCBE can be used for the clinical treatment of mitochondrial diseases. Here, we focused on human pathogenic mtDNA mutations, and conducted this proof of concept study that a screened DdCBE could mediate base editing to correct homoplasmic m.A4300G mutation with limited by-stander editing and off-target editing. Our data showed that the phenotypes including mitochondrial protein level and mitochondrial respiration were restored by repairing m.A4300G mutation back to wild-type allele at over 60%, confirming that the impaired mitochondrial function is truly caused by this mutation. Our results provide reference for developing any gene-editing based therapy strategies

for targeting m.A4300G mutation in the future.

1. The authors use plasmid to deliver DdCBE into cells, so the risk of RNA off-target in iPSC induced by DdCBE (including deaminase) should be evaluated. Random integration also should be tested.

Thanks for the insightful suggestion. To exclude the integration of DdCBE in iPSC genome, we analyzed 25 edited iPSC clones and did not detect any integration of each DdCBE monomer in all these iPSC clones (G1397C and G1397N, new Supplementary Fig.4a), which was also demonstrated by the whole genome sequence analysis of G4300A-#54, #81 and #84 (data not shown). Please find Line 192-194: *As a result, these clones harbored G4300A conversion with frequencies that ranged from 1.14 to 72.97% (Fig.2f and Supplementary Fig.3d) and without integration of DdCBE pair in the genome (Supplementary Fig.4a).*

Based on our analysis of DdCBE integration, we hold the opinion that the risk of RNA off-target is negligible in edited iPSC clones. As a deaminase, DddA_{tox} we used in this study, only catalyzes the deamination of cytosine within double-stranded DNA, rather than within single-stranded DNA, single-stranded RNA and double-stranded RNA (PMID: 32641830). In addition, DdCBE pairs were not integrated into iPSCs genome, which means the expression of DdCBE is transient in the 25 iPSC clones. Herein, even if there is any possibility that DdCBE could potentially induce off-targets on RNA, this off-target effect would be diminished after several passages in these corrected iPSC clones.

new Supplementary Fig.4a

a. Detection of the integrated DddAtox halves in the genome of genetically corrected iPSC clones.

2. Line 41, the author applied DdCBE with different TALE pair to target mitochondrial DNA, technically, it is not optimized, it is screened or selected. “An optimized DdCBE pair” seems like “the whole DdCBE system has been upgraded”.

Thanks for your suggestion. We have revised it as “a screened DdCBE pair” in the main text.

3. Cells after gene editing need to be tested for pluripotency and karyotype.

Thanks for your valuable suggestions. We tested the pluripotency and karyotypes of two edited iPSC clones, both of which displayed normal karyotypes and expressed pluripotent markers (new Supplementary Fig. 4d to 4f). Please find Line 208-209: Importantly, the edited iPSC clones still had normal karyotypes as expected and expressed pluripotent markers (Supplementary Fig.4d-4f).

new Supplementary Fig.4d to 4f

d. qRT-PCR verification of pluripotency markers NANOG, OCT4 and LIN28 in G4300A-#54, #84 and amniocytes. Data are presented as mean \pm SEM; n = 3 independent experiments.

e. The karyotype of G4300A-#54 and #84.

f. Immunofluorescence staining of pluripotency markers SSEA4, TRA-1-60 and TRA-1-81 in G4300A-#54 and #84. Scale bars, 50 μ m.

4. In Fig.4e, COX1 and COX2 protein levels seems different from Fig1. Completing quantitative analysis would be better.

Thanks for your suggestion. We determined the protein expression with three independent repeats and completed the quantitative analysis for the five cell lines (new Supplementary Fig.S6d). The COX1 and COX2 protein levels in contrl and A4300G-iPSCs are similar with Fig. 1. One thing we have to be honest about is that, although

the COX2 level of A4300G-iPSCs was decreased compared with control, we did not detect a significant difference between the two groups. This lack of significance may be attributed to the variations in cell confluence among different batches.

new Supplementary Fig.6d

d. The quantitative analysis of COX1, COX2 and COX3 protein level in control, m.A4300G-iPSCs and corrected iPSC clones. Data are presented as mean \pm SEM, $n = 3$ independent experiments.

5. Flow cytometry figures about iPSCs transfection efficiency should be provided.

Thanks for your valuable suggestion. We have provided the flow cytometry figures about iPSCs transfection efficiency in the main text (Line 190) and new Supplementary Fig.3c. The proportion of double positive cells were 6.3% in total nucleofected cells and cells with stronger fluorescence signal (purple dots) were collected for further study.

Supplementary Fig.3c

c

c. Flow cytometry of untreated iPSCs and nucleofected iPSCs. Purple dots indicate the sorted cells for culture.

Reviewer #3 (Remarks to the Author):

This manuscript presented a patient-derived induced pluripotent stem cell (iPSC) line generated from a pathological specimen, aiming to address the m.A4300G mutation through the application of Dd-CBE. The ensuing experiment effectively demonstrated the notable benefits of high gene editing efficiency and reduced bystander effects. Nevertheless, several questions remain.

We thank reviewer for the positive remarks.

1. Primarily based on the fundamentals of Dd-CBE, the gene editing strategy employed in this study raises the inquiry of whether any novel elements were introduced. Additionally, considering the relative frequency of Dd-CBE usage in iPSC research, it is crucial to ascertain any advancements compared to prior investigations.

Thanks for the valuable suggestion. In this study, we screened canonical DdCBE pairs with best performance to precisely correct the m.A4300G mutation and minimize the bystander editing through optimizing the TALE array, spacer region and combinations of the DddA_{tox} splits, without introducing any additional novel elements. However, based on our current data, several key issues still need to be addressed for gene therapy, including bystander editing and off-target editing. Recent advances in mtDNA editing such as the development of HiFi-DdCBEs could be employed to decrease off-target editing and unwanted bystander editing (PMID: 36229610). In discussion section, we added these information and proposed strategies to further optimize DdCBE for gene therapy (Line 326-338, We propose several strategies to further optimize DdCBE for mitochondrial gene therapy: 1) Use evolved DddAs and DddA homologs to enhance DdCBE's activity and expand its targeting scope on mtDNA; 2) Fuse transactivators (VP63, P65 or Rta) to the end of UGI to improve the frequencies of C-to-T conversions; 3) Substitute DddAtox with high-fidelity DddA to reduce mitochondrial off-targets; 4) Introduce nuclear export signal (NES) to the DdCBE system or simultaneously express DddIA in the nucleus to reduce nuclear off-targets; 5) DNA strand-selective

mitochondrial base editors could be an alternative strategy to reduce bystander editing. These strategies may help obtain genetically corrected iPSC clones more efficiently and precisely. Additionally, our study suggests considering several factors when utilizing DdCBE, including the number of tC motif within the spacing region, the length of the spacing region, the contexts of TALE binding sites, and the orientation and split types of DddAtox. All of these factors collectively contribute to the performance of DdCBE pairs.

To the best of our knowledge, none previous work has already been performed using DdCBE in iPSC research. Compared to previous studies on gene therapy using mito-TALEN, ZFN and allotopic overexpression of mitochondrial proteins in cell lines, including iPSCs, the advancements of DdCBE in our manuscript are capable of correcting homoplasmic mutations on both mitochondrial coding and non-coding genes in situ, which were emphasized in our original introduction (Line 61 to 78).

2. Regarding the therapeutic strategy associated with this iPSC gene editing pipeline, an essential aspect to clarify is whether the objective is to utilize edited iPSCs directly or to derive cardiomyocytes from these edited iPSCs. Is it feasible to directly edit the patient's cardiomyocytes? Substantial evidence is still required to establish the viability of employing Dd-CBE in gene therapy.

Thanks for the valuable suggestion. Our objective is to conduct a proof of concept study that whether DdCBE could mediate correction of human homoplasmic mutations on mtDNA of patient derived iPSCs. For this purpose, we performed our research mainly on iPSCs directly, because it is a human cell model carrying the pathogenic mtDNA mutation, which is proper to answer the raised question. As we know, mitochondrial diseases are very complicated with clinically heterogeneous, no one knows whether damaged mitochondrial function can be rescued by genetic correction, and what proportion of mtDNA mutations corrected can rescue the phenotype. More importantly, preclinical studies are required before DdCBE can be used for the clinical treatment of mitochondrial diseases. Here, we focused on human pathogenic mtDNA mutations, and

conducted this proof of concept study that a screened DdCBE could mediate base editing to correct homoplasmic m.A4300G mutation with limited by-stander editing and off-target editing. Our data showed that the phenotypes including mitochondrial protein level and mitochondrial respiration were restored by repairing m.A4300G mutation back to wild-type allele at over 60%. Our results provide reference for developing any gene-editing based therapy strategies for targeting this pathogenic mutation in the future.

Regarding the feasibility to directly edit the patient's cardiomyocytes, previous report demonstrated that DdCBE successfully installed mutants on mtDNA by delivering DdCBEs into wildtype mouse heart using adeno-associated virus (AAV) vectors (PMID: 35136065), suggesting the possible feasibility to directly edit the patient's cardiomyocytes. However, such work has not been done on human cardiomyocytes yet.

Yes, we totally agree with the reviewer's opinion that substantial evidence is still required to establish the viability of employing DdCBE in gene therapy. Our work is merely a small preliminary step in this field, and substantial further research is still required to advance the early clinical application of this technology in treating patients with mitochondrial diseases. We have added the relating description in the discussion. Please find Line 311-325: *In terms of the development of related therapies, it has been demonstrated that DdCBE can install mutations on mtDNA by delivering DdCBEs into wild-type mouse heart using AAV vectors, suggesting the possible feasibility to directly edit the patient's cardiomyocytes. However, mitochondrial diseases are extremely complicated with clinically heterogeneous. It is still unknown whether damaged mitochondrial function can be restored through genetic correction and what proportion of mtDNA mutations corrected can rescue the phenotype. More importantly, preclinical studies using patient-derived iPSCs are necessary before DdCBE can be used for the clinical treatment of mitochondrial diseases. In this study, we focused on human pathogenic mtDNA mutations and conducted a proof of concept study to demonstrate that a screened DdCBE could mediate base editing to correct the homoplasmic m.A4300G mutation, thus restoring impaired mitochondrial function in patient-derived*

iPSCs. It is important to note that our work represents a small preliminary step in this field, and substantial further research is still required to advance the early clinical application of this technology for treating patients with mitochondrial diseases.

3. Apart from gene editing efficiency and off-target effects, are there any other factors that necessitate consideration when utilizing Dd-CBE? Considering the distinct characteristics of each patient, should Dd-CBE pair screening still be conducted? Furthermore, what specific attributes contribute to the successful performance of Dd-CBE pairs?

Thanks for your insightful comments. This is almost an emerging field, and we believe there are many issues that need to be considered before clinical use, such as how to avoid bystander editing, whether there is pre-existing immunity against therapeutic DdCBE protein, etc.

In addition, it still needs to consider the distinct characteristics of each patient. For example, each patient carries distinct SNPs, it needs to adjust the TALE sequence accordingly, because alterations in TALE sequence may also affect the efficiency of DdCBE. Hence, we think it necessary to perform DdCBE pair screening for each individual.

Our data suggested that the length of spacer region, orientation and split types of DdCBE all contribute to the performance of DdCBE pairs. Based on this and our previous studies (PMID: 37058569, 34938607, 35102135, 34663794, 34480028), we did not find any specific attributes that determine the performance of DdCBE, suggesting the necessity of DdCBE pairs screening.

We have added the statements in the discussion. Please find Line 311-338: *In terms of the development of related therapies, it has been demonstrated that DdCBE can install mutations on mtDNA by delivering DdCBEs into wild-type mouse heart using AAV vectors, suggesting the possible feasibility to directly edit the patient's cardiomyocytes. However, mitochondrial diseases are extremely complicated with clinically*

heterogeneous. It is still unknown whether damaged mitochondrial function can be restored through genetic correction and what proportion of mtDNA mutations corrected can rescue the phenotype. More importantly, preclinical studies using patient-derived iPSCs are necessary before DdCBE can be used for the clinical treatment of mitochondrial diseases. In this study, we focused on human pathogenic mtDNA mutations and conducted a proof of concept study to demonstrate that a screened DdCBE could mediate base editing to correct the homoplasmic m.A4300G mutation, thus restoring impaired mitochondrial function in patient-derived iPSCs. It is important to note that our work represents a small preliminary step in this field, and substantial further research is still required to advance the early clinical application of this technology for treating patients with mitochondrial diseases.

We propose several strategies to further optimize DdCBE for mitochondrial gene therapy: 1) Use evolved DddAs and Ddda homologs to enhance DdCBE's activity and expand its targeting scope on mtDNA; 2) Fuse transactivators (VP63, P65 or Rta) to the end of UGI to improve the frequencies of C-to-T conversions; 3) Substitute DddAtox with high-fidelity Ddda to reduce mitochondrial off-targets; 4) Introduce nuclear export signal (NES) to the DdCBE system or simultaneously express DddIA in the nucleus to reduce nuclear off-targets; 5) DNA strand-selective mitochondrial base editors could be an alternative strategy to reduce bystander editing. These strategies may help obtain genetically corrected iPSC clones more efficiently and precisely. Additionally, our study suggests considering several factors when utilizing DdCBE, including the number of tC motif within the spacing region, the length of the spacing region, the contexts of TALE binding sites, and the orientation and split types of DddAtox. All of these factors collectively contribute to the performance of DdCBE pairs.

REVIEWERS' COMMENTS:

Reviewer #1 (Remarks to the Author):

All of my comments were fixed. I recommend to publish the current format to Communications Biology.

Reviewer #2 (Remarks to the Author):

The authors have satisfactorily addressed my concerns.

Reviewer #3 (Remarks to the Author):

I have gone through the responses and found they basically addressed all my comments. No more questions.